# Trends and Determinants of Cigarette Tax Increases in Japan: The Role of Revenue Targeting

**DOI:** 10.3390/ijerph19084892

**Published:** 2022-04-18

**Authors:** Takashi Oshio, Ryota Nakamura

**Affiliations:** 1Institute of Economic Research, Hitotsubashi University, Tokyo 186-8603, Japan; 2Hitotsubashi Institute for Advanced Study, Hitotsubashi University, Tokyo 186-8601, Japan; ryota.nakamura@r.hit-u.ac.jp

**Keywords:** cigarette sales, cigarette tax, cigarette tax revenue, Ministry of Finance

## Abstract

Cigarette prices in Japan are lower than those in most other high-income countries. A more striking fact is that cigarette tax revenues have been kept almost flat at just over two trillion JPY (Japanese yen; 18.2 billion US dollars) over more than three decades, despite steadily declining cigarette sales and seemingly weakening pressure from stakeholders with a vested interest in the tobacco industry. We attempted to examine trends and determinants of cigarette tax increases in Japan. In particular, we hypothesized that the Japanese finance ministry adjusts cigarette taxes to meet a revenue target. Under this hypothesis, we searched for the most plausible amount of the minimum target of tax revenue that corresponds to cigarette tax increases over the past 37 years (1985–2021) using public data on cigarette sales and taxes. The results revealed that two trillion JPY was the minimal revenue target that could plausibly explain the increase in cigarette tax. In addition, the timing and magnitude of cigarette tax increases have been successfully set to maintain stable tax revenues. A key determinant of cigarette tax increases in Japan has been hard revenue targets, rather than public health concerns.

## 1. Introduction

Despite a series of cigarette tax increases, cigarette prices remain much lower in Japan than in other countries. Among 38 member countries in the Organisation of Economic Development and Co-operation (OECD), the price of a standard 20-cigarette pack of the most sold brand in Japan (in US dollars adjusted for purchasing power) was the second-lowest among the OECD member countries, and around one-fourth of the highest country (New Zealand) [1]. Along with relatively loose tobacco control measures, lower cigarette prices have often been criticized for preventing the prevalence of smokers from declining smoothly [2,3,4].

A well-established, political economic view emphasizes the tobacco industry-government interaction [5]. In the Japanese context, such a view reflects pro-tobacco legislators, regulatory oversight of the primary Japanese tobacco company, Japan Tobacco Inc. (JT), from the Ministry of Finance (MOF), and industry interference on the policy-making process, putting a key barrier to effective tobacco control measures and cigarette price increases in Japan [2,6,7,8,9,10], as is the case in other countries [11,12]. There are pockets of political resistance in the National Diet to higher cigarette taxes and tighter restrictions on smoking [6,7,8,9], as represented in the Alliance of LDP Smokers in the ruling Liberal Democratic Party (LDP). As a result, public trust in government tobacco control remained low [13].

However, the landscape of the tobacco industry-government interaction shows changes over the past decades, reflecting a substantial reduction in the number of smokers and cigarette sales. The adult smoking rate has decreased from 60.5% in men and 14.3% in women in 1990 to 27.8% and 8.7% in 2018 [14], and the number of cigarettes sold annually fell from 348.3 billion in 1996 to 98.8 billion in 2020 [15]. If heated cigarette products are added, the latest sales figures were a little higher—about 140 billion in 2020—but still 60% below the historical peak [16]. Underlying this downturn is a public climate increasingly hostile to smoking, as seen in the growing number of local governments imposing smoking bans and restrictions to protect the public from secondhand smoke, although the effects of these policies have been ineffective [17,18].

Reflecting a reduction in cigarette sales, the number of domestic cigarette farmers has plummeted from 78,653 in 1985 to 4340 in 2020 [16]. The number of retailers has also declined to 237,000 in 2019 after peaking at 307,000 in 2001, with convenience stores replacing once-ubiquitous neighborhood cigarette shops and taking over the bulk of retail activity [16]. As a result, the number of stakeholders with a vested interest in the tobacco industry seems to have declined to the point where political pressure can no longer be as influential as before.

Amid this seemingly inexorable decline in smoking and cigarette sales, the stability of cigarette tax revenues over the years is striking. Since 1985, when JT took over the industry from the previous government monopoly, Japan Cigarette and Salt Public Corporation, the combined revenue from national and local cigarette taxes has rarely fallen below two trillion JPY (Japanese yen; 18.2 billion US dollars). As a result, the proportion of cigarette tax revenues in total tax revenue has been hovering in the 2–3% range [19,20]. A combination of declining cigarette sales and stable cigarette tax revenues over the past three decades has fueled the notion that the makers of Japanese tax policy are determined to raise the cigarette tax to maintain the annual revenue of two trillion JPY [21], amid the seemingly declining political pressure from the tobacco industry and related stakeholders.

Despite the fact that the smoking rate has decreased gradually and cigarette tax rates have increased, albeit modestly compared to other OECD countries, tobacco policies in Japan lack inter-ministerial coordination between the MOF and Ministry of Health, Labour and Welfare (MHLW). The MOF exclusively determines cigarette tax rates; while cigarette taxes consist of roughly equally sized national and local components, the MOF determines both components and controls the total cigarette tax revenue. In contrast, the MHLW plays essentially no role when the MOF sets cigarette tax rates. There is no health promotion fund in the tax structure, and tax revenues are not earmarked in the government expenditures.

As a result, the cigarette tax has not been considered a public health policy measure but has merely been an important source of tax revenue. This could be one of the important reasons to explain why cigarette prices in Japan are the second-lowest among the OECD member countries.

Hence, it is crucial to investigate the mechanism under which cigarette taxes have been determined over time. We believe that this study is one of the first attempts to formally address this issue in the relevant literature. Using the data on cigarette sales and taxes, we examined how the MOF has been responding to an expected change (reduction) in cigarette sales. Specifically, we explored the most plausible amount of the minimum target of total cigarette tax revenue—the sum of national and local tax revenues—that would trigger cigarette tax increases. Based on the observation of cigarette tax revenue, which has remained stable at just above two trillion JPY over more than three decades, we hypothesized that any drop in the revenue forecast below two trillion JPY would trigger a cigarette tax increase. We also examined the extent to which a projected increase in cigarette taxes could fill the projected revenue shortage.

## 2. Materials and Methods

### 2.1. Data

The present study used annual data on cigarette tax rates, tax revenue and sales between 1985 and 2020 from different, publicly available sources. The cigarette tax rate was calculated using the amount of tax per piece of cigarette, of which data were obtained from the Ministry of Finance statistics monthly and the MOF’s website [19]. The data on cigarette tax revenues were also obtained from the MOF [19]. In Japan, cigarette tax is collected locally (the prefecture level) as well as centrally (the national level). This study used the sum of national and local tax revenues in the analyses throughout. Finally, the data on cigarette sales were obtained from the Tobacco Institute of Japan, an association of tobacco manufacturers, distributors and other stakeholders in Japan [15]. The sales data include the total number of pieces of cigarettes sold in the country annually.

### 2.2. Statistical Analysis

We considered two simple regression models, Models 1 and 2, in which any drop in the revenue forecast below a certain value, *X* trillion JPY, would trigger a cigarette tax increase. We assumed that the MOF projected cigarette sales in volume to increase in year *t*, *E*(*V_t_*), from the previous year at a trend pace—which was tentatively assumed to be given by a pace averaged over the last five years—if there is no change in cigarette taxes (and prices) per unit, that is, *E*(*V_t_*) = *V*_*t*−1_ × (*V*_*t*−1_/*V*_*t*−6_)^1/5^. We also constructed a binary variable of the expected revenue shortage, *I* (*X* − *τ*_*t*−1_*E*(*V_t_*) > 0), by allocating one to the case in which *X* − *τ*_*t*−1_*E*(*V_t_*) > 0 and zero otherwise, where the average tax per cigarette, *τ*, was assumed to be fixed from year *t* − 1. 

We considered two models, Models 1 and 2, to describe the MOF’s behavior. Model 1 explains a binary variable of cigarette tax increase (*Taxhike*) by binary variables of the expected revenue shortage, and an increase in consumption tax (*CThike*), which likely affected a cigarette tax increase. Hence, Model 1 is given by:$$\mathrm{Model}\;1:\;Taxhike_{t}=\alpha_{1}+\beta_{1}I\left[X-\tau_{t-1}E\left(V_{t}\right)>0\right]+\gamma_{1}CThike_{t}+\varepsilon_{1t},$$
where CThike is a binary variable of a consumption tax increase, and *ε*_1_ is an error term. 

In Model 2, we replaced a binary variable of the expected revenue shortage by its amount:$$\mathrm{Model}\;2:\;Taxhike_{t}=\alpha_{2}+\beta_{2}\mathrm{Max}\left[X-\tau_{t-1}E\left(V_{t}\right),0\right]+\gamma_{2}CThike_{t}+\varepsilon_{2t},$$
which implies that the higher the expected revenue shortage, the higher the probability of the tax increase. *ε*_2_ is an error term.

By raising the value of *X* from 1.5 to 2.5 (trillion JPY) at intervals of 0.1, we repeatedly estimated Models 1 and 2, respectively, for each value of *X*. Then, we searched for a value of *X* that maximizes the goodness of fit of each model. In these estimations, we first applied the linear probability model (LPM) analysis [22,23], which linearly regresses the binary dependent variable on the explanatory variables, and searched for a value of *X* that maximizes the goodness of fit of the model in terms of adjusted *R*-squared. Then, we replaced the LPM with a probit model and searched for a value of *X* that maximizes the goodness of fit of the model in terms of log likelihood. 

Based on the most plausible value of the minimum tax revenue target, we further examined the extent to which an increase in cigarette tax revenue projected by the model could fill the expected shortage of tax revenue. Given the price elasticity of demand for cigarettes assumed by the MOF, *e*, we estimated a linear regression model, Model 3, which is given by
$$\mathrm{Model}\;3:\left(1-e\Delta \tau_{t}/\tau_{t-1}\right)E\left(V_{t}\right)\Delta \tau_{t}=\alpha_{3}+\beta_{3}\mathrm{Max}\left[X-\tau_{t-1}E\left(V_{t}\right),0\right]+\varepsilon_{3t},$$
where Δ*τ* indicates a planned increase in the tax per cigarette, the left-hand side of the equation indicates a projected increase in cigarette tax revenue, and *ε*_3_ is an error term. One study showed that the price elasticity (absolute value) of demand for cigarettes ranged between 0.27 and 0.30 [24], but we did not know how the MOF actually assumed its value. Hence, we searched for a value of *e* that maximized the goodness of fit of this equation in terms of the adjusted *R*-squared by raising the value of *e* from 0 to 1 at intervals of 0.1. 

It should be noted the regression analysis in this study ignored macroeconomic or social factors that may potentially affect people’s demand for cigarettes or the MOF’s taxation, which may confound the relationship between the tax rate and revenue. Hence, the estimation results indicate a statistical correlation rather than any causal mechanism. The Stata software package (Release 17, StataCorp., College Station, TX, USA) was used for all of the statistical analyses.

## 3. Results

Figure 1 describes the recent trends of cigarette sales (which included heated tobacco products in 2017 and later) and cigarette tax per unit, based on statistics published by the Japan Tobacco Institute and MOF, respectively. As seen in this figure, cigarette sales steadily declined since the late 1990s, while the cigarette tax was steadily raised, with a jump in 2010. These two offsetting factors probably contributed to stable tax revenues. In fact, Figure 2 shows that cigarette tax revenue (the sum of national and local taxes) was maintained to mark slightly above two trillion JPY over decades. Figure 2 also indicates that tax increases (independent of the consumption tax increases) tended to be conducted one year after cigarette tax revenues dropped below JPY two trillion or were about to break that threshold, except for the cases in 2003 and 2006. Notably, a recent series of tax increases in 2018–2021 seem to be closely related to a continuous revenue shortfall of JPY two trillion.

Appendix A summarizes the simulation results of Models 1 and 2 by focusing the response of the goodness of fit to different values of the minimum target of cigarette tax revenue, *X*. All models showed that the goodness of fit was maximized when *X* was equal to 2 (trillion JPY). We repeated the calculations by raising the value of *X* at intervals of 0.01 (rather than 0.1), and obtained the optimal values of *X* as 1.97 (LPM-Model 1), 2.00 (LPM-Model 2), 1.97 (Probit-Model 1), and 1.98 (Probit-Model 2), all of which were close to 2 (not reported in the table). We also observed that the goodness of fit was somewhat better in Model 2 than in Model 1 for both the LPM and probit models. This result implies that the MOF may have been more concerned about the magnitude of the projected tax revenue shortfall than the risk of the shortfall.

Table 1 summarizes the estimation results of Models 1 and 2 when the value of *X* is set to 2. The LPM results showed that a projected shortfall of the tax revenue would raise the probability of the tax increase by 46.8% in Model 1 and that a 10 billion JPY shortage of tax revenue raises the probability by 4.5%. A coefficient of *CThike* was positive but not significant, suggesting the cigarette tax increases that were largely independent of the consumption tax increase. Probit regressions also confirmed a close correlation between the projected shortage of tax revenue and cigarette tax increases.

For Model 3, in which the value of *X* is assumed as 2, we found that the goodness of fit of the model was maximized when the price elasticity of demand for cigarettes, *e*, was equal to zero, implying that the MOF may have assumed that consumers cannot necessarily adjust their demand for cigarettes to price changes in the short run. Assuming that *X* = 2 and *e* = 0, Table 2 provides the estimation results of Model 3. The coefficient of the expected shortage of tax revenue (0.869) was close to one, and the constant term (0.017) was close to zero. Indeed, the joint hypothesis that the coefficient of the expected shortage of the tax revenue was equal to one and that the constant term was zero cannot be rejected (*p* = 0.455). This result suggests that the MOF has been delicately adjusting cigarette taxes to avoid the tax revenue from falling short of the target, even if the Ministry was not concerned about smokers’ short-term response to tobacco price increases.

These results demonstrated that the MOF’s strategy was best explained by the minimum revenue threshold of two trillion JPY, but it did not necessarily mean that a reduction in the cigarette tax revenue led to a cigarette tax increase. As a supplementary analysis, we conducted a time series analysis of the cigarette tax data to address this issue. The results statistically confirmed that a reduction in the expected tax revenue would lead to an increase in tax rate after one or two years (see Appendix A).

## 4. Discussion

Using the data released by the MOF and Tobacco Institute of Japan, we examined how the timing and magnitude of the cigarette tax increases were associated with a target of cigarette tax revenues, which have remained stable at approximately a little more than two trillion JPY over more than three decades. Our statistical analysis showed that the two trillion JPY targets could explain the pattern of cigarette tax increases in the past. The MOF tended to raise cigarette taxes if it expected cigarette tax revenue to fall short of two trillion JPY. This decision might hinge more on the margin of the revenue shortage than the existence of any anticipated shortfall. Moreover, the magnitude of the tax increase is consistent with the revenue shortfall projected by the MOF.

Apart from political pressure related to the tobacco industry, this study highlights the importance of the government’s motivation to maintain tax revenue for cigarette tax policies in Japan. To be sure, we did not identify any obvious reason for the MOF to maintain the target of two trillion JPY. However, the results of this study suggest that this revenue target has been so binding that the government’s cigarette tax policies have been endogenously affected by the trend of cigarette consumption. Our supplementary analysis confirmed that a reduction in expected cigarette tax revenue would lead to an increase in the tax rate after one or two years.

The results also suggest that cigarette tax increases have been delicately adjusted to avoid tax revenue from falling short of the target. This method of maintaining the tax revenue seems to account for lower cigarette prices compared to other high-income countries, which in turn may lead to a lower rate of reduction in the adult smoking rate in Japan. Another noticeable finding was that the models best fit the data when we adopted a scenario in which the MOF assumed zero price elasticity of demand for cigarettes at least in the short run, a result consistent with a well-established knowledge about the demand for addictive goods.

The results underscore the need of inter-ministerial collaboration between the ministries of finance and health. Even though the MHLW has advocated the national agenda for health promotion calling for smoking prevention, our analysis showed that cigarette tax increases are largely explained as a response to declining tobacco sales, not public health concerns. It is, however, well known that the annual societal costs attributable to smoking—in terms of health impacts, lost productivity, healthcare expenditures, etc.—far exceed two trillion JPY [25]. Given the societal costs of cigarette smoking, it is hard to justify dependence on the cigarette industry as a means of financing government expenditures. However, the results suggest that societal costs are not a concern of the tax authorities.

As suggested in the broad literature (e.g., refs. [26,27,28]), a cigarette tax hike would reduce cigarette consumption and a reduction in consumption would reduce tax revenue, motivating the government to increase the tax rate to maintain tax revenue in the short term. The gradual increase in the cigarette tax rate over decades in Japan appears to be best explained by this cyclic process. As the MHLW have implemented a series of policy interventions to prevent smoking, the decline in smoking is expected to accelerate, and as demand plummets, further tax increases by the MOF will be needed to secure tax revenue. Hence, taxes and cigarette prices could spiral upward as demand spirals downward, regardless of arguably weakening pressure from the tobacco industry. This means that, even though the Japanese government does not raise the cigarette tax to promote public health, smoking and its societal costs will continue to decline. It should be noted, however, that smoking is more prevalent and entrenched among populations that are socioeconomically disadvantaged with respect to factors such as education, income, and employment [29,30,31]. It means that the economic impact of higher cigarette taxes falls disproportionately on those who can least afford it.

Hence, there is a need for tighter restrictions and prohibitions on smoking in public places and more effective measures to deny minors access to cigarettes. We also need to provide more support and encouragement to smokers who want to quit smoking. Outpatient smoking cessation programs, which are already covered by health insurance, should be more widely available. Companies could contribute by offering special perks to employees who quit smoking. Research in behavioral economics may yield hints as to the most effective ways to help people quit smoking.

As an implication for future studies on the tobacco tax, our study highlighted the endogeneity of the cigarette tax policy in Japan, in contrast with many studies that have considered the policy exogenously and examined its impact on the demand for cigarettes and smoking behavior [4,24,26,32,33]. In particular, this study presented the first empirical analysis on the determinants of cigarette tax increases in Japan, while one study indicated that prefectures involved in growing tobacco exhibited lower levels of compliance with national tobacco control laws [34].

This study has several limitations. First, more detailed information is required about the decision-making process of cigarette taxation policy and its relationship with non-tax policies to examine how explicitly and seriously the two trillion JPY target has been considered by policymakers. Second, we assumed no change in demand for cigarettes in response to tax increases, at least in the short term, and ignored the possibility that higher taxes may change a smoker’s brand preference. Third, we did not consider how the dividend income earned by the MOF, which owns approximately 33% of JT shares, has affected the cigarette tax policy, although the Ministry’s dividend income (70–100 billion yen annually) remained around 5% or less of the cigarette tax revenue.

## 5. Conclusions

In this study, we showed that a decline in the cigarette tax revenue is a crucial determinant of the tobacco tax increase in Japan. We argued that, with an implicit targeting of tobacco tax revenue, over the past decades the MOF increased the tax rate when the revenue was expected to fall below the target. This contrasts with the conventional public health view, in which the government increases cigarette tax to reduce tobacco consumption.

The findings of the study should be interpreted with caution. Notably, the MOF’s internal process for taxation decisions are not directly observed and hence are at best inferred from the data. Furthermore, the analyses were based on the assumptions of no structural changes in smokers’ behavior, and also on no consideration of the impact of cigarette tax revenues on other public policies.

However, the results of this study suggest that a key determinant of cigarette tax increases in Japan is a hard revenue target, rather than public health concerns. This will likely continue to hold despite weakening pressure from the tobacco industry, and it should cause a further spiral between rising cigarette taxes and declining cigarette sales in the foreseeable future. The disproportionate impact of increasing cigarette taxes on smokers of lower socioeconomic status is a new issue to be addressed in tobacco control.

Tobacco policies in Japan should enhance inter-ministerial coordination between the MOF and MHLW. The MOF could play an important role to reflect public health in the cigarette tax policy [35], whereas the MHLW should enhance its commitment to the decision-making process of cigarette taxation. Strong political leadership will be needed to strengthen such inter-ministerial coordination to reduce smoking rates further and promote public health.

## Figures and Tables

**Figure 1 ijerph-19-04892-f001:**
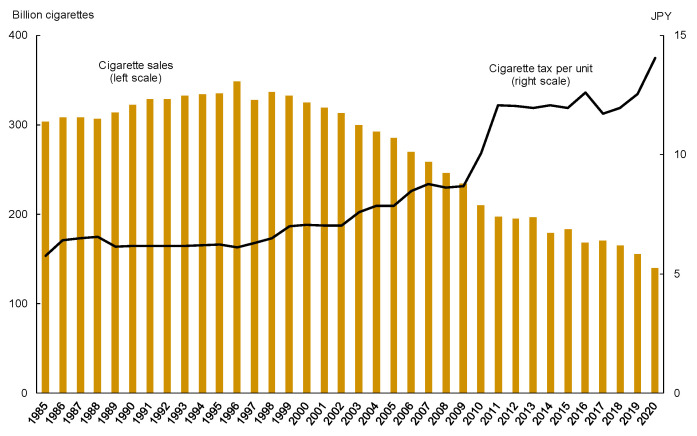
Trends in cigarette sales and cigarette tax per unit.

**Figure 2 ijerph-19-04892-f002:**
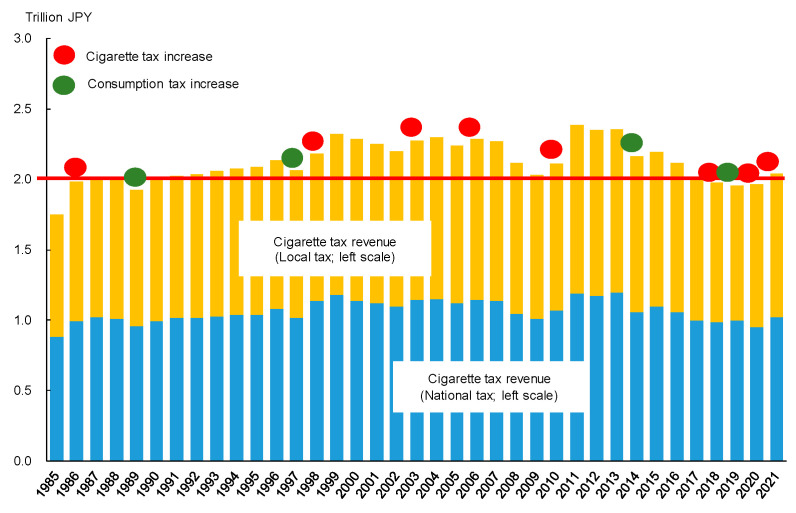
Trend in cigarette tax revenue and timings of cigarette tax increases.

**Table 1 ijerph-19-04892-t001:** Estimation results of models to explain the probability of a cigarette tax increase ^a^.

**(1) Linear Probability Model (LPM)**		**Coef.**	**95% CI ^b^**	**Adjusted *R*^2^**	** *N* **
Model 1	*I*[2 − *τ_t_*_−1_ *E*(*V_t_*) > 0]	0.468 **	(0.141, 0.795)	0.204	36
	*CThike*	0.351	(−0.161, 0.864)		
	Constant	0.133	(−0.008, 0.327)		
Model 2	Max[2 − *τ_t_*_−1_ *E*(*V_t_*), 0]	4.633 ***	(2.01, 7.26)	0.280	36
	*CThike*	0.741 ***	(0.36, 1.12)		
	Constant	0.176 *	(0.03, 0.33)		
**(2) Probit model**		**Coef.**	**95% CI**	**Log-likelihood**	** *N* **
Model 1	*I*[2 − *τ_t_*_−1_ *E*(*V_t_*) > 0]	1.36 **	(0.32, 2.39)	−17.755	36
	*CThike*	1.12	(–0.56, 2.80)		
	Constant	−1.01 ***	(–1.60, −0.42)		
Model 2	Max[2 − *τ_t_*_−1_ *E*(*V_t_*), 0]	25.14 *	(4.04, 46.25)	−15.296	36
	*CThike*	1.07	(–0.73, 2.86)		
	Constant	–1.05	(–1.63, −0.47)		

^a^ Assuming that the minimum target of cigarette tax revenue was two trillion JPY. ^b^ Confidence interval. *** *p* < 0.001, ** *p* < 0.01, * *p* < 0.05.

**Table 2 ijerph-19-04892-t002:** Estimation results of models to explain a projected increase in cigarette tax revenue ^a^.

OLS		Coef.	95% CI ^b^	Adjusted *R*^2^	*N*
Model 3	Max[2 − *τ_t_*_−1_ *E*(*V_t_*), 0]	0.869 ***	(0.399, 1.339)	0.272	36
	Constant	0.017	(−0.008, 0.042)		

^a^ Assuming that the minimum target of cigarette tax revenue was two trillion JPY and that the price elasticity of demand for cigarettes was zero. ^b^ Confidence interval. *** *p* < 0.001.

## Data Availability

Publicly available datasets were analyzed in this study. This data can be found here: http://www.health-net.or.jp/tobacco/product/pd090000.html; https://www.tioj.or.jp/data/pdf/210531_04.pdf (accessed on 12 April 2022); and https://www.mof.go.jp/tax_policy/summary/consumption/d09.htm (accessed on 12 April 2022).

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
