# Peer review of "Trends and Determinants of Cigarette Tax Increases in Japan: The Role of Revenue Targeting"

_ijerph, 2022, doi:10.3390/ijerph19084892_

Round 1
Reviewer 1 Report
The aim is very hard, and it could be introduce noise on public health and governal activities of Japan on tobacco taxes.
I suggest to modify some sentence for example in the abstract you heve to use more diplomatic way: "We hypothesized that the Japanese finance ministry manipulates cigarette taxes to meet a hard revenue target."
I suggest to use a caution on the Conclutions of the work.
Author Response
Thank you very much for the opportunity to revise the manuscript. We addressed all the comments, which we found very helpful to improve the manuscript. In our responses we refer to line numbers of the revised manuscript with “track changes” (that is, the numbers include deleted lines).
Reviewer 1:
The aim is very hard, and it could be introduce noise on public health and governal activities of Japan on tobacco taxes.
I suggest to modify some sentence for example in the abstract you have to use more diplomatic way: “We hypothesized that the Japanese finance ministry manipulates cigarette taxes to meet a hard revenue target.”
=> In response to the comments, we have made the expression “more diplomatic” by revising the sentence as the following. “We hypothesized that the Japanese finance ministry adjusts cigarette taxes to meet a revenue target” (line 13). We also replaced “manipulate” with “adjusted” wherever appropriate, and removed accusatory sentences (as suggested by Reviewer 2).
I suggest to use a caution on the Conclutions of the work.
=> We have added a remark for a cautious interpretation of the work to Conclusion, based on the limitations mentioned in Discussion: “The findings of the study should be interpreted with caution. Notably, the MOF’s internal process for taxation decisions are not directly observed and hence are at best inferred from the data. Furthermore, the analyses were based on the assumptions of no structural change in smokers’ behavior, and also on no consideration about the impact of cigarette tax revenues on other public policies.” (lines 301-305).
Reviewer 2 Report
IJERPH - 2022 – 1616078 Referee Report
In this paper, the authors examine cigarette prices in Japan, which are lower than those of other advanced countries. The authors note that while cigarette prices in Japan are low, tax revenues have been kept constant over more than three decades at around the two trillion Japanese Yen threshold. Given that no public information is available to explain this revenue threshold, the authors hypothesize that the Japanese finance ministry simply manipulates cigarette taxes to meet a revenue target, instead of using changes in taxes to discourage cigarette use. The authors then use 37 years (1985-2021) of publicly available data on cigarette sales and taxes to model the most plausible amount of minimum target of tax revenue that would trigger a chance in cigarette taxes.
The results of this paper contribute to the literature of taxation and prices of e-cigarettes. This study is mildly interesting, and I would strongly suggest the authors work on the discussion to better solidify the importance of this study. As written the discussion doesn’t provide policy recommendations that may dissuade the Japanese MOF to focus on public health instead of revenue targets.
Furthermore, I would recommend the following revisions:
Introduction:
- The paragraph found on lines 78-86 is better suited for the conclusion. I would suggest moving this paragraph to the conclusion and use it as a paragraph that summarizes the study.
Methods:
- Figure 1 has both local taxes and a national tax. This is the first instance that a reader learns about local taxes, which are then never mentioned again in the manuscript. Does the MOF have control over local taxes? How does this affect the authors theory? This needs to be explored in more detail.
- Lines 173 - 175 – the authors mention that the price elasticity of demand that maximizes the goodness of fit of the model is equal to zero. I think the interpretation the authors provide is slightly erroneous. A price elasticity of demand equal to zero implies that demand for a good will not change with changes in prices – this is especially true in the short run when consumers cannot adjust to a change in prices. While the MOF may not be concerned about a possible reduction in the demand for cigarettes in the short run, it is probably truer that the MOF knows consumers can not necessarily adjust in the short run to changes in price of an addicting good.
Discussion:
- Line 214, I believe the authors meant to write: Since cigarette taxes have been a reliable and steady source of tax revenue, Japanese government authorities would like to keep it that way…
- The paragraph found on lines 214-219 is hard to follow. I suggest the authors either delete it, or re-word it. As it currently stands, it is hard to understand what the authors are trying to convey.
- The sentence on line 230 is to casual and accusatory for a journal article, this type of wording is best kept in editorials and should be removed or reworded.
- One of the most important findings of this paper is a price elasticity of demand equal to zero in the short run, the authors should highlight this finding and elaborate further on the implications of this finding. This finding is in line with economic theory and the authors should capitalize on this finding.
Mistakes:
- Figure 1 footnotes have a typo, as it has two footnote a’s. One should be removed. It is also hard to find where these footnotes are in the manuscript. To that end, having the footnotes on lines 184 and 185 seems out of place.
Author Response
Thank you very much for the opportunity to revise the manuscript. We addressed all the comments, which we found very helpful to improve the manuscript. In our responses we refer to line numbers of the revised manuscript with “track changes” (that is, the numbers include deleted lines).
Reviewer 2:
In this paper, the authors examine cigarette prices in Japan, which are lower than those of other advanced countries. The authors note that while cigarette prices in Japan are low, tax revenues have been kept constant over more than three decades at around the two trillion Japanese Yen threshold. Given that no public information is available to explain this revenue threshold, the authors hypothesize that the Japanese finance ministry simply manipulates cigarette taxes to meet a revenue target, instead of using changes in taxes to discourage cigarette use. The authors then use 37 years (1985-2021) of publicly available data on cigarette sales and taxes to model the most plausible amount of minimum target of tax revenue that would trigger a chance in cigarette taxes.
The results of this paper contribute to the literature of taxation and prices of e-cigarettes. This study is mildly interesting, and I would strongly suggest the authors work on the discussion to better solidify the importance of this study. As written the discussion doesn’t provide policy recommendations that may dissuade the Japanese MOF to focus on public health instead of revenue targets.
=> In response to the constructive comments, we have added the following sentences to Conclusion: “Tobacco policies in Japan should enhance inter-ministerial coordination between the MOF and MHLW. The MOF could play an important role to reflect public health in the cigarette tax policy [35], whereas the MHLW should enhance its commitment to the decision-making process of cigarette taxation. Strong political leadership will be needed to strengthen such an inter-ministerial coordination to reduce smoking rates further and promote public health” (lines 312-317).
Furthermore, I would recommend the following revisions:
Introduction:
The paragraph found on lines 78-86 is better suited for the conclusion. I would suggest moving this paragraph to the conclusion and use it as a paragraph that summarizes the study.
=> We have moved the sentence to Conclusion (295-300). We have also removed the references to previous studies in the US considering the context.
Methods:
Figure 1 has both local taxes and a national tax. This is the first instance that a reader learns about local taxes, which are then never mentioned again in the manuscript. Does the MOF have control over local taxes? How does this affect the authors theory? This needs to be explored in more detail.
=> We have added the explanation: “while cigarette taxes consist of roughly equally sized national and local components, the MOF determines both components and controls the total cigarette tax revenue” (lines 74-76). In addition, we have made it clearer that the MOF aims to control the sum of cigarette tax revenues at both levels, by stating “we explored the most plausible amount of the minimum target of total cigarette tax revenue – the sum of national and local tax revenues –that would trigger cigarette tax increases” (lines 88-90). Division between national and local cigarette tax revenues does not affect our theory.
Lines 173 - 175 – the authors mention that the price elasticity of demand that maximizes the goodness of fit of the model is equal to zero. I think the interpretation the authors provide is slightly erroneous. A price elasticity of demand equal to zero implies that demand for a good will not change with changes in prices – this is especially true in the short run when consumers cannot adjust to a change in prices. While the MOF may not be concerned about a possible reduction in the demand for cigarettes in the short run, it is probably truer that the MOF knows consumers can not necessarily adjust in the short run to changes in price of an addicting good.
=> We have replaced our interpretation following the reviewer’s suggestion by stating: “implying that the MOF may have assumed that consumers cannot necessarily adjust their demand for cigarettes in the short run to price changes” (lines 191-192).
Discussion:
Line 214, I believe the authors meant to write: Since cigarette taxes have been a reliable and steady source of tax revenue, Japanese government authorities would like to keep it that way…
=> We have removed the whole paragraph including this sentence, as suggested by the reviewer’s following comment.
The paragraph found on lines 214-219 is hard to follow. I suggest the authors either delete it, or re-word it. As it currently stands, it is hard to understand what the authors are trying to convey.
=> We have removed the paragraph.
The sentence on line 230 is to casual and accusatory for a journal article, this type of wording is best kept in editorials and should be removed or reworded.
=> We have removed the sentence.
One of the most important findings of this paper is a price elasticity of demand equal to zero in the short run, the authors should highlight this finding and elaborate further on the implications of this finding. This finding is in line with economic theory and the authors should capitalize on this finding.
=> We appreciate the reviewer’s suggestion. We have added the following sentence to Discussion: “Another noticeable finding was that the models best fit the data when we adopted a scenario in which the MOF assumed zero price elasticity of demand for cigarette at least in the short run, a result consistent with a well-established knowledge about the demand for addictive goods.” (lines 235-238).
Mistakes:
Figure 1 footnotes have a typo, as it has two footnote a’s. One should be removed. It is also hard to find where these footnotes are in the manuscript. To that end, having the footnotes on lines 184 and 185 seems out of place.
We have removed ‘a’ and notes from Figure 1 and included the contents in the footnotes in the manuscript (lines 153-155).
Reviewer 3 Report
WHO FCTC Article 6 recommends that the Parties of the FCTC (including Japan) increase tobacco tax. I believe the Japanese government has taken actions based on the FCTC and has successfully decreased tobacco consumption in Japan. I see that there has been a rapid reduction in cigarette sales between 1998 and 2020. MOF is not a department for health, so it's obvious that they increased tobacco tax based on their tax revenue target.
Rather than the authors' current argument, I think how the Japanese government uses the tobacco tax revenue is a more interesting issue. If the government spend that money on other places rather than tobacco control policy or smoking cessation service, that would be a problem.
What is the structure of the cigarette tax in Japan? Is there any health promotion fund in the tax structure? What is the role of the Health Department when MOF increase tobacco tax?
I don't see any new findings and am not sure what the authors wanted to present in this paper.
Author Response
Thank you very much for the opportunity to revise the manuscript. We addressed all the comments, which we found very helpful to improve the manuscript. In our responses we refer to line numbers of the revised manuscript with “track changes” (that is, the numbers include deleted lines).
Reviewer 3:
WHO FCTC Article 6 recommends that the Parties of the FCTC (including Japan) increase tobacco tax. I believe the Japanese government has taken actions based on the FCTC and has successfully decreased tobacco consumption in Japan. I see that there has been a rapid reduction in cigarette sales between 1998 and 2020. MOF is not a department for health, so it’s obvious that they increased tobacco tax based on their tax revenue target.
Rather than the authors’ current argument, I think how the Japanese government uses the tobacco tax revenue is a more interesting issue. If the government spend that money on other places rather than tobacco control policy or smoking cessation service, that would be a problem.
=> Thank you very much for this comment. We do not deny the fact that smoking rate has decreased gradually and tobacco tax has increased (yet modestly as compared to other OECD countries) in Japan. We argue, however, that the tobacco tax has not been considered nor implemented as a public health measure in the Japanese context, which shows a stark contrast with other more successful countries in raising tobacco tax and reducing smoking rate. This could be one of important reasons why the price of cigarette remained second lowest in OECD countries. We claim that this study is the first one to formally address this issue in the relevant literature.
We have made this point clearer in Introduction by stating: “Despite the fact that the smoking rate has decreased gradually and cigarette tax rates have increased, albeit modestly compared to other OECD countries, tobacco policies in Japan lack inter-ministerial coordination between the MOF and Ministry of Health, Labour and Welfare (MHLW). The MOF exclusively determines cigarette tax rates; while cigarette taxes consist of roughly equally sized national and local components, the MOF determines both components and controls total cigarette tax revenues. In contrast, the MHLW plays essentially no role when the MOF sets cigarette tax rates. There is no health promotion fund in the tax structure, and tax revenues are not earmarked in the government expenditure. / As a result, cigarette tax has not been considered a public health policy measure but has merely been an important source of tax revenue. This could be one of important reasons to explain why cigarette prices in Japan are the second-lowest among the OECD member countries. / Hence, it is crucial to investigate the mechanism under which cigarette taxes have been determined over time. We believe that this study is one of the first attempts to formally address this issue in the relevant literature” (lines 71-86).
What is the structure of the cigarette tax in Japan? Is there any health promotion fund in the tax structure? What is the role of the Health Department when MOF increase tobacco tax?
=> We have added the following explanation to Introduction: “The MOF exclusively determines cigarette tax rates; while cigarette taxes consist of roughly equally sized national and local components, the MOF determines both components and controls total cigarette tax revenues. In contrast, the MHLW plays essentially no role when the MOF sets cigarette tax rates. There is no health promotion fund in the tax structure and tax revenues are not earmarked in the government expenditure” (lines 74-79).
I don’t see any new findings and am not sure what the authors wanted to present in this paper.
=> We have made our argument clearer in Conclusion by adding: “In this study, we demonstrated that a decline in the cigarette tax revenue is a crucial determinant of the tobacco tax increase in Japan. We argued that, with an implicit targeting of tobacco tax revenue, over the past decades the MOF increased the tax rate when the revenue was expected to fall below the target. This contrasts with the conventional public health view, in which the government increases cigarette tax to reduce tobacco consumption” (lines 295-300).